# Ribonucleic Acid Export 1 Is a Kinetochore-Associated Protein That Participates in Chromosome Alignment in Mouse Oocytes

**DOI:** 10.3390/ijms22094841

**Published:** 2021-05-03

**Authors:** Fan Chen, Xiao-Fei Jiao, Fei Meng, Yong-Sheng Wang, Zhi-Ming Ding, Yi-Liang Miao, Jia-Jun Xiong, Li-Jun Huo

**Affiliations:** 1Key Laboratory of Agricultural Animal Genetics, Breeding and Reproduction of Ministry of Education, College of Animal Science and Technology, Huazhong Agricultural University, Wuhan 430070, China; fanny151110@yahoo.com (F.C.); xiaofei.jiao@yahoo.com (X.-F.J.); MengFei@webmail.hzau.edu.cn (F.M.); wang1035233950@yahoo.com (Y.-S.W.); dingzhiming10@163.com (Z.-M.D.); miaoyl@mail.hzau.edu.cn (Y.-L.M.); xiongjiajun@mail.hzau.edu.cn (J.-J.X.); 2Department of Hubei Province Engineering Research Center in Buffalo Breeding and Products, Wuhan 430070, China

**Keywords:** mouse oocyte, meiotic maturation, Rae1, chromosome alignment, aneuploidy, securin

## Abstract

Ribonucleic acid export 1 (Rae1) is an important nucleoporin that participates in mRNA export during the interphase of higher eukaryotes and regulates the mitotic cell cycle. In this study, small RNA interference technology was used to knockdown Rae1, and immunofluorescence, immunoblotting, and chromosome spreading were used to study the role of Rae1 in mouse oocyte meiotic maturation. We found that Rae1 is a crucial regulator of meiotic maturation of mouse oocytes. After the resumption of meiosis (GVBD), Rae1 was concentrated on the kinetochore structure. The knockdown of Rae1 by a specific siRNA inhibited GVBD progression at 2 h, finally leading to a decreased 14 h polar body extrusion (PBE) rate. However, a comparable 14 h PBE rate was found in the control, and the Rae1 knockdown groups that had already undergone GVBD. Furthermore, we found elevated PBE after 9.5 h in the Rae1 knockdown oocytes. Further analysis revealed that Rae1 depletion significantly decreased the protein level of securin. In addition, we detected weakened kinetochore–microtubule (K-MT) attachments, misaligned chromosomes, and an increased incidence of aneuploidy in the Rae1 knockdown oocytes. Collectively, we propose that Rae1 modulates securin protein levels, which contribute to chromosome alignment, K-MT attachments, and aneuploidy in meiosis.

## 1. Introduction

The nuclear pore complex (NPC) is a massive multiprotein complex that is embedded in the nuclear envelope and regulates the nucleocytoplasmic transport of diverse molecules in the eukaryotic interphase. A single NPC is roughly composed of 30 different protein components termed nucleoporins [1]. Rae1 (homologs in *Schizosaccharomyces pombe*, Rae1p; *Saccharomyces cerevisiae*, Gle2p) was initially identified in *S. pombe* as a nucleoporin involved in poly(A) mRNA export [2,3]. Subsequent studies have shown that in addition to mRNA export, Rae1 participates in mitotic progression [4]. Rae1 harbors a microtubule–associated activity in higher eukaryotes. In *Xenopus* egg extracts, the Rae1–RNA ribonucleoprotein complex controls microtubule polymerization [5]. In HeLa cells, Rae1 interacts with the nuclear mitotic apparatus protein (NuMA), and this interaction and balance is critically required for normal bipolar spindle assembly [6]. A study in U2OS cells validated that ubiquitin-specific protease 11 (USP11) binds to and modulates the ubiquitination of Rae1, thereby regulating the interaction with NuMA, which is critical for normal bipolar spindle formation [7]. In addition, Rae1 interacts with chromosome subunit 1 (SMC1), and disrupting this interaction can result in spindle abnormalities in HeLa cells [8,9].

In higher eukaryotes, Rae1 shares an extensive sequence homology with the mitotic checkpoint protein Bub3 and can interact with Bub1 [10,11]. Further analysis has indicated that the haploinsufficiency of either Rae1 or Bub3 induces a similar phenotype involving mitotic checkpoint defects and chromosome mis-segregation, and the overexpression of Rae1 could correct defects of both Rae1 haploinsufficiency and Bub3 haploinsufficiency in mouse embryonic fibroblast (MEF) lines, which indicates that Rae1 may be analogous to Bub3 [12]. All of the above evidence suggests that Rae1 serves as a mitotic checkpoint regulator.

As an important nucleoporin, Rae1 has been extensively studied in mRNA transport and mitotic processes. Rae1 is also critically required for male meiosis and spermatogenesis [13]; however, whether Rae1 exerts a regulatory function during oocyte meiosis is still unexplored. Compared with mitosis, oocyte meiosis is a unique kind of cell division that couples two successive chromosome divisions with only one round of DNA replication to yield a highly polarized egg. Moreover, oocytes undergo protracted arrest at the diplotene stage of prophase I during meiosis [14]. Oocyte meiosis tends to result in chromosome segregation errors, which are especially high in humans, and chromosome mis-segregation is highly correlated with the generation of aneuploidy, which is a leading factor of recurrent spontaneous abortion and congenital defects [15]. Faithful chromosome segregation relies on stable connections between kinetochores and spindle microtubules. To avoid false chromosome separation and aneuploidy, the mammalian oocyte has a surveillance pathway to monitor the kinetochore–microtubule (K-MT) attachments, the spindle assemble checkpoint (SAC), which is composed of Bub and Mad family members [16,17]. The SAC can discriminate between correct and incorrect K-MT attachments and inhibit the activity of the anaphase-promoting complex or cyclosome (APC/C) E3 ubiquitin ligase activity to generate a “stop anaphase” signal until all incorrect attachments are eventually corrected [18].

In our previous study, we investigated the function of another nucleoporin, Nup35, in mouse oocytes. After the resumption of meiosis, Nup35 shows a dynamic spindle location and participates in spindle assembly [19]. However, unlike Nup35, Rae1 is concentrated on the kinetochore structure during meiosis I. Moreover, Rae1–RNAi oocytes showed defects in chromosome alignment, K-MT attachments and aneuploidy, which may be caused by decreased securin expression levels.

## 2. Results

### 2.1. Subcellular Localization and Expression Pattern of Rae1 during Mouse Oocyte Maturation

We first investigated the subcellular localization of Rae1 at different meiotic maturation stages by immunofluorescence staining. As shown in Figure 1A, the immunofluorescence signal of Rae1 was concentrated on the nuclear rim at the germinal vesicle (GV) stage, which is consistent with previous results [6,12]. Moreover, shortly after meiotic resumption, Rae1 was concentrated on the chromosome structure. By further chromosome spreading, we verified that Rae1 accumulated on the kinetochore structure of the chromosome and colocalized with the kinetochore complex CREST (Figure 1B,C). The kinetochore localization of Rae1 indicates its specific role during oocyte meiotic maturation. Furthermore, the relative protein level of Rae1 was examined using Western blotting (Figure 1D). Rae1 had comparable protein levels at the GV, GVBD (germinal vesicle breakdown), and metaphase I (MI) stages, but had the highest level at the metaphase II (MII) stage (Figure 1E).

### 2.2. Rae1 Knockdown Weakens Meiotic Resumption but Precociously Promotes the First Polar Extrusion 

For further functional analysis, Rae1 was knocked down by the microinjection of Rae1-specific siRNA into GV stage oocytes. Western blotting results verified that the relative level of Rae1 at the GV stage was significantly decreased in the Rae1–siRNA microinjected oocytes (Figure 2A). The knockdown efficiency of Rae1 was also validated with the immunostaining signal of Rae1 in the GV stage oocytes (Figure 2B). These results indicated that Rae1–RNAi achieved a good knockdown efficiency worthy of further study. We then evaluated the effect of Rae1 knockdown on oocyte meiotic maturation. The results showed that the knockdown of Rae1 significantly affected the GVBD rate compared with that in the control group (87.3 ± 2.7% vs. 70.4 ± 5.0%, *p* < 0.05, Figure 2C). The rate of polar body extrusion (PBE) at 14 h significantly decreased in the Rae1 knockdown oocytes (63.7 ± 2.1% vs. 45.7 ± 3.9%, *p* < 0.05, Figure 2D); however, the proportion of oocytes with a polar body was comparable in the control and Rae1–RNAi meiosis-resumed oocytes (already undergoing GVBD; 79.0 ± 3.1% vs. 65.4 ± 5.4%, *p* >0.05, Figure 2E). Moreover, the PBE rate after 9 h was significantly advanced in the Rae1 knockdown group (14.3 ± 3.0% vs. 25.9 ± 2.0%, *p* < 0.05, Figure 2F), indicating a precocious anaphase trigger. These results showed that Rae1 participates in the GVBD and PBE processes.

### 2.3. Rae1 Interference Induces Abnormal Chromosome Alignment and a High Incidence of Aneuploidy

As Rae1 participated in the meiotic maturation process and localized to the kinetochore, we assessed the spindle morphology and chromosome alignment in the MI stage oocytes. The results showed that oocytes had the typical meiotic spindle morphology in both the control and RNAi groups. The chromosomes were well aligned at the bipolar spindle plate in the control oocytes; however, dispersed chromosomes were observed in the Rae1–RNAi oocytes (Figure 3A). The rate of misaligned chromosomes was significantly elevated in the RNAi group compared with the control group (19.3 ± 3.3% vs. 39.5 ± 2.6%, *p* < 0.01, Figure 3B). Given that misaligned chromosomes at the MI stage are a leading factor of aneuploidy [20], we then examined the aneuploidy in MII oocytes by chromosome spreading, and the results showed that the aneuploid rate was also significantly increased in the Rae1 knockdown oocytes (22.3 ± 2.6% vs. 42.0 ± 3.6%, *p* < 0.01, Figure 3C,D). These results suggested that Rae1 is involved in chromosome alignment at the MI stage and faithful chromosome segregation at the transition of metaphase I to anaphase I.

### 2.4. Rae1 Depletion Compromises K-MT Attachments during Oocyte Meiotic Maturation

Faithful chromosome alignment and segregation also depend on accurate and stable K-MT attachments [21]. Therefore, we used cold treatment to assess the K-MT attachment; following low temperature treatment, accurate and strong K-MT attachment remained stable, whereas erroneous and fragile K-MT attachment was disassembled [22]. In this study, MI oocytes were collected after cold treatment and were immunostained for the kinetochore proteins CREST and α-tubulin. In the control group, we observed that CREST was evenly positioned on both sides of the center of the spindle, and microtubules mostly captured the kinetochores on the aligned bivalents on the spindle equator. Conversely, we found that the CREST location pattern had some deviations, and we detected a higher frequency of unattached kinetochores in Rae1 knockdown oocytes (Figure 4A), indicating that some kinetochores were weakly attached to microtubules after Rae1 knockdown. The rate of unattached kinetochores was significantly higher (20.3 ± 2.9% vs. 62.3 ± 5.7%, *p* < 0.01; Figure 4B) in the Rae1 knockdown oocytes than in the control oocytes. These results indicate that Rae1 is crucial for proper K-MT attachment and accurate chromosome alignment.

### 2.5. Rae1 Depletion Had No Effects on BubR1 and Mad1 Location

SAC inhibits the metaphase–anaphase transition until the chromosomes are correctly aligned and K-MT attachments are accurately connected [18]. Given the observed defects in chromosome alignment, aneuploidy, and K-MT attachment, we investigated the SAC function in the control and Rae1 RNAi groups. The key SAC components, including BubR1 and Mad1, were assessed in the control and Rae1 knockdown oocytes cultured for 8.5 h after release from IBMX block. As shown in Figure 5A,B, the immunofluorescence signals of both BubR1 and Mad1 normally persisted at the kinetochore in the control and Rae1 knockdown oocytes. The results suggested that the activity of SAC was unaffected by Rae1 knockdown.

### 2.6. Rae1 Depletion Decreases the Protein Level of securin but Not Cyclin B

Anaphase-promoting complex or cyclosome (APC/C) directly controls the metaphase–anaphase transition by regulating the proper degradation of securin and cyclin B [21]. In this experiment, we assessed the protein levels of securin and cyclin B in the control and Rae1 knockdown oocytes at the MI stage. As shown in Figure 6, the protein level of securin was significantly decreased in the Rae1 knockdown oocytes; however, the protein levels of both cyclin B1 and cyclin B2 were similar between the control oocytes and the Rae1 knockdown oocytes, which indicated that Rae1 specifically modulates the protein level of securin.

## 3. Discussion

Previous observations have shown that Rae1 serves as a nuclear exporter of poly(A) mRNA at interphase, and a microtubule-associated protein and mitotic checkpoint regulator at mitosis. In this study, we investigated the role of Rae1 in mouse oocytes. We found that Rae1 concentrates on the nuclear rim in the GV stage and translocates to the kinetochore structure after meiotic resumption. Rae1 knockdown induces decreased securin protein levels, accompanied by misaligned chromosomes, aberrant K-MT attachments, and a high rate of aneuploidy.

A high intraoocyte level of cAMP activates protein kinase A (PKA), which in turn inhibits the activity of maturation-promoting factor (MPF), maintaining oocytes in the diplotene stage [23]. MPF is a heterodimer composed of the catalytic subunit CDK1 and regulatory subunit cyclin B [24]. In this study, we found that the GVBD rate was significantly decreased in the Rae1 knockdown oocytes; however, we did not observe significant changes in the cyclin B1 and cyclin B2 levels. Therefore, we concluded that the knockdown of Rae1 has no influence on MPF activity. The disassembly of NPCs is a decisive event during nuclear membrane breakdown (NEBD). Nup98 is the first component to dissociate from the nuclear envelope upon NEBD, and the phosphorylation of Nup98 is a key step in NPC disassembly [25]. Rae1 and Nup98 exist and function together as subcomplexes of Rae1/Nup98 during the interphase and mitotic phases [26,27]. Therefore, the knockdown of Rae1 may affect the dissociation of Nup98 from the nuclear envelope, which eventually interferes with the GVBD process.

After examining the expression pattern of Rae1 during oocyte meiotic maturation, we found that the Rae1 protein level in the oocytes was lower in the MI stage, but much higher in the metaphase II (MII) stage. Rae1 knockdown significantly decreased the securin level, resulting in a precocious anaphase trigger and aneuploidy. Published data have validated that homologous chromosome separation at the transition of metaphase I to anaphase I depends on the proteolysis of securin [28]. In contrast, the MII stage is maintained by the securin protein. MII oocytes from aged mice have lower securin levels and a higher rate of aneuploidy than those from young mice [29]. Therefore, we speculate that the lower level of Rae1 in MI oocytes might be related to the transition from metaphase I to anaphase I (securin proteolysis), and that the higher level of Rae1 in MII oocytes might be related to the maintenance of MII stage arrest.

In both mitosis and meiosis, SAC monitors chromosome alignment and K-MT attachments, and persists at the kinetochore to inhibit the precocious anaphase trigger until proper chromosome alignment and K-MT connections are corrected [30]. In this study, Rae1 knockdown induced precocious anaphase, misaligned chromosomes, and a high rate of aneuploidy, which conformed to previous data [12,31]. All these results showed that the activity of SAC might be impaired. However, in our results, we did not detect any different localization patterns of BubR1 and Mad1 (main component of SAC) between the control and Rae1 knockdown groups, which indicates that SAC functions normally after Rae1 knockdown, and we asssume the precocious anaphase may be caused by the decreased securin level.

SAC can inhibit APC/C E3 ubiquitin ligase activity to generate a “stop anaphase” signal until incorrect attachments eventually become corrected [18]. Once the chromosomes are correctly aligned and SAC dissociates from the kinetochore, activated APC/C promotes the degradation of securin and cyclin B and triggers anaphase [32]. In this study, we observed decreased protein levels of securin but not Cyclin B in the Rae1 knockdown oocytes. Consistently, published data have also shown that Rae1/Nup98 can specifically inhibit APC^Cdh1^-mediated ubiquitination and the degradation of securin in prometaphase [27,32]. However, during the metaphase–anaphase transition, another nucleoporin, Nup88, can sequester Rae1/Nup98 away from APC^Cdh1^, which promotes the degradation of securin [33]. In a word, Nup88 and Rae1/Nup98 composes of a regulatory network that regulates the activity of APC^Cdh1^ to control the securin level, thus regulating chromosome alignment and anaphase onset.

## 4. Materials and Methods

### 4.1. Animal Statement

Kunming mice, a native breed widely used in biological research, were used for oocyte collection and experiments in this study. Four- to six-week-old female Kunming mice were purchased from the animal center of Huazhong Agricultural University. All experimental procedures and animal treatments were performed in accordance with the rules stipulated by the Animal Care and Use Committee of Huazhong Agricultural University (HZAUSW-2017-005).

### 4.2. Antibodies and Reagents

Mouse anti-Rae1 monoclonal antibody (Cat# SC-393252), mouse anti-Mad1 monoclonal antibody (Cat# 376613), rabbit anti-Cyclin B2 antibody (Cat# sc-2776), and mouse anti-securin monoclonal antibody (Cat# SC-56207) were obtained from Santa Cruz Biotechnology (Santa Cruz, CA, USA); mouse anti-α-tubulin-FITC antibody (Cat# F2168) was obtained from Sigma Chemical Company (St Louis, MO, USA); mouse anti-α-tubulin monoclonal antibody (66031-1-Ig) and mouse anti-β-actin monoclonal antibody (66009-1-Ig) were bought from Proteintech (Wuhan, China); rabbit anti-Cyclin B1 antibody (Cat# AF6168) was purchased from Affinity (Cincinnati, OH, USA); human anti-CREST antibody (Cat# 15-234-0001) was purchased from Antibodies Incorporated (Davis, CA, USA); sheep anti-BubR1 polyclonal antibody (Cat# 28193) was obtained from Abcam (Cambridge, UK); and DyLight 549-conjugated goat anti-mouse IgG (H + L) was purchased from Abbkine Biotechnology (California, CA, USA). FITC-conjugated donkey anti-sheep IgG (H + L) was produced by Jackson ImmunoResearch Laboratory (West Grove, PA, USA), FITC-conjugated goat anti-human IgG (H + L) was purchased from Boster (Wuhan, China), and TRITC-conjugated goat anti-human IgG (H + L) was purchased from Proteintech (Wuhan, China). All other chemicals and culture media were purchased from Sigma Chemical Company (St Louis, MO, USA), unless otherwise stated.

### 4.3. Oocyte Retrieval and Culture

Four- to six-week-old female Kunming mice were sacrificed by cervical dislocation 48 h after intraperitoneal injection of 5 IU pregnant mare serum gonadotropin (PMSG; Solarbio, P9970). Fully grown GV stage oocytes were collected by ovarian puncture in a prewarmed F12 medium (Ham, YC-3034) supplemented with 50 μM IBMX (Sigma, 15879, St Louis, MO, USA), and were then used for culture or microinjection. For meiotic maturation, oocytes were washed three times to remove IBMX and then cultured in a prewarmed MII (Sigma, M7167) medium covered with paraffin oil (Sigma, M8410) at 37 °C in a 5% CO_2_ atmosphere. Different stages of oocytes were collected for the following experiments.

### 4.4. Microinjection of Rae1-Targeted Short Interfering siRNA

Eppendorf CellTram oil was used for the microinjection. After isolation from the ovary, fully grown GV stage oocytes were used for immediate microinjection in MII medium supplemented with 50 μM IBMX. The microinjection operation of 200 oocytes should be rapid and within 1 h, with no significant effect on the subsequent developmental potential of oocytes. For Rae1 knockdown, 5–10 pl of 50 μM Rae1-targeted short interfering siRNA (Cat# sc-75826; Santa Cruz, CA, USA) was microinjected into the cytoplasm of fully grown GV stage oocytes, and the same amount of negative control siRNA (Cat# sc-37007; Santa Cruz, CA, USA) was used as a control. After microinjection, the oocytes were arrested at the GV stage in an MII medium containing 50 μM IBMX for 24 h to degrade the targeted mRNA. Then, the oocytes were washed three times to remove IBMX and were cultured in a prewarmed MII medium for an additional 14 h.

### 4.5. Immunofluorescence and Confocal Microscopy

Oocytes at different stages were collected and fixed in 4% paraformaldehyde (in PBS, pH 7.4; Sigma, 158127) for 30 min and permeabilized with 0.5% Triton-X-100 (Biosharp, 0649) in PBS for 30 min at room temperature. Then, the oocytes were blocked with a blocking buffer (2% BSA-supplemented in PBS; Sigma, A2153) for 1 h and incubated with an anti-Rae1 antibody (1:100), anti-α-tubulin-FITC antibody (1:100), or anti-CREST antibody (1:200) at 4 °C overnight. After three washes in a washing solution (1% Tween 20 and 0.01% Triton-X 100 in PBS), the oocytes were incubated with DyLight 549-conjugated goat anti-mouse (1:100) or TRITC-conjugated goat anti-human (1:50) for 1 h at room temperature. After three washes in a washing solution, the oocytes were counterstained with DAPI (Invitrogen, D130610, 1:1000, Waltham, MA, USA) for 1–5 min. Finally, the oocytes were mounted and observed under a confocal laser scanning microscope (Carl Zeiss 800, Jena, Thüringen, Germany). Confocal images were processed using Zeiss LSM Image Browser software. Oocytes incubated with fluorescence-labelled secondary antibodies were used as a negative control, and the primary antibody was replaced with nonimmune IgG.

### 4.6. Western Blotting 

Samples containing 200 oocytes were collected at the GV, GVBD, MI, and MII stages in a 2X SDS loading buffer and were boiled for 5 min. After separation by SDS-PAGE, the proteins were transferred to polyvinylidene fluoride membranes (Sigma, CCGL09025). After transfer, the membranes were briefly washed and blocked in a blocking buffer (5% skim milk in TBST) at room temperature for 1 h, followed by incubation overnight at 4 °C with a Rae1 antibody (1:400), securin antibody (1:400), cyclin B1 antibody (1:500), or cyclin B2 antibody (1:500) at 4 °C overnight. After three washes in TBST, the membranes were incubated at 37 °C for 1 h with HRP-conjugated secondary antibodies. Finally, the membranes were washed in TBST, and the immunoblot bands were visualized with an ECL kit (Thermo Scientific, A38555, Waltham, MA, USA). α-tubulin antibody (1:1000) and β-actin (1:1000) served as the loading controls. The relative intensity of the band was assessed using ImageJ software (NIH, Bethesda, MD, USA). A blank control incubated with an HRP-conjugated secondary antibody was used as the negative control, and the primary antibody was replaced by nonimmune IgG.

### 4.7. Oocyte Cold Treatment

Fully grown GV stage oocytes microinjected with either the control siRNA or Rae1 siRNA were maintained in an MII medium containing 50 μM IBMX for 24 h, then the IBMX was washed off and the oocytes were cultured in a fresh MII medium. Two hours later, the GV stage oocytes were removed, and the remainder were cultured for another 6 h. The oocytes (MI stage) were then transferred to a precooled MII medium (4 °C) for 15 min of incubation. After cold treatment, immunofluorescence staining and confocal microscopy were used to detect the immunofluorescence signal of CREST and others. Oocytes with normal spindles in both the control and RNAi groups were used to compare the K-MT attachment.

### 4.8. Chromosome Spreading

Chromosome spreading was performed as described previously [34]. Briefly, oocytes were collected in the MI stage and were exposed to Tyrode’s buffer (pH 2.5, Sigma, T1788) for approximately 30 s at room temperature to remove the zona pellucida. After recovery in an MII medium for 30 min, the oocytes were fixed in a drop of spreading solution (1% PFA, 0.15% Triton X-100, 3 mM DTT in ddH2O, pH 9.2) on a glass slide. After air drying, the slides were washed and blocked with 1% BSA in PBS, followed by incubation with anti-Rae1 antibody (1:100), anti-CREST antibody (1:200), anti-BubR1 antibody (1:100), or anti-Mad1 antibody (1:40) at 4 °C overnight. Next, the primary antibody was washed away, and the oocytes were incubated with DyLight 549-conjugated goat anti-mouse (1:100) antibody, FITC-conjugated goat anti-human (1:100) antibody, or FITC-conjugated donkey anti-sheep antibody (1:100) at 37 °C for 1 h. The chromosomes were stained with DAPI. Images of the immunofluorescence staining of oocytes were captured with confocal microscopy.

### 4.9. Statistical Analysis

All of the experiments were independently performed at least three times; data are presented as the mean ± SEM and were analyzed with two-tailed Student’s t test using GraphPad software (CA, USA); *p* < 0.05 was considered significant.

## 5. Conclusions

In conclusion, this study demonstrated that Rae1 concentrates on the kinetochore after the resumption of meiosis and regulates the securin protein level to regulate K-MT attachment and faithful chromosome alignment. Rae1 might not function alone, but instead interacts with other nucleoporins, including Nup88 and Nup98.

## Figures and Tables

**Figure 1 ijms-22-04841-f001:**
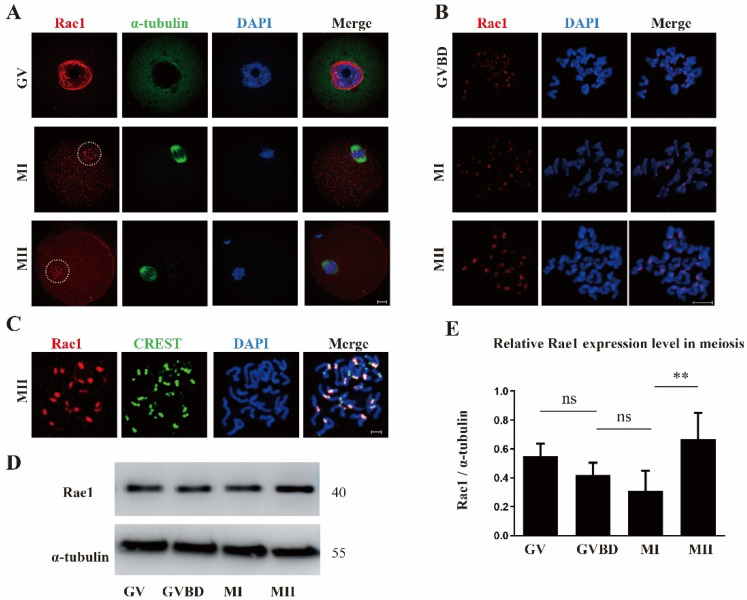
The subcellular localization and protein expression pattern of Rae1 during oocyte meiotic maturation. (**A**) Representative images of the subcellular localization of Rae1 in oocyte at the germinal vesicle (GV), metaphase I (MI), and metaphase II (MII) stages. Rae1 is shown in red, α-tubulin in green, and 4’,6-diamidino-2-phenylindole (DAPI) in blue. Scale bar = 10 μm. (**B**) Representative images of Rae1 localization in chromosomes of oocytes at the germinal vesicle breakdown (GVBD), MI, and MII stage by chromosome spreading. Rae1 is shown in red and DAPI in blue. Scale bar = 5 μm. (**C**). The co-localization of Rae1 and CREST in chromosomes of the MIIstage oocyte by chromosome spreading. Rae1 is shown in red, CREST in green, and DAPI in blue. Scale bar = 5 μm. (**D**) The protein level of Rae1 at the GV, GVBD, MI, and MII stages detected by Western blotting. (**E**) Relative intensity of Rae1 at the GV, GVBD, MI, and MII stages is presented, which were calculated by Rae1/tubulin. NS means no significant difference. ** *p* < 0.01. The data are presented as mean ± standard error of the mean (SEM) of three independent experiments.

**Figure 2 ijms-22-04841-f002:**
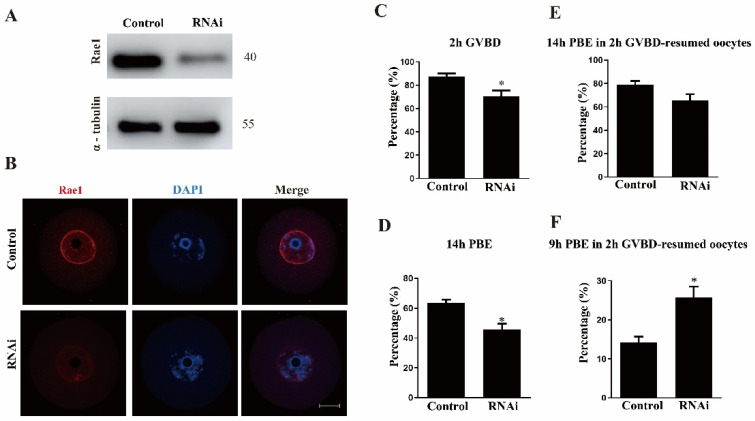
Rae1 knockdown affects the meiotic maturation process. (**A**) Efficiency of Rae1–RNAi after siRNA microinjection was verified by immunoblotting. (**B**) RNAi after siRNA microinjection was verified by immunostaining. Rae1 is shown in red and DAPI in blue. Scale bar = 20 μm. (**C**) The GVBD rates in the control group and Rae1 knockdown group were recorded 2 h after release from 3-Isobutyl-1-Methylxanthine (IBMX) arrest. * *p* < 0.05. (**D**). The PBE rates in the control group and Rae1 knockdown group were recorded 14 h after release from IBMX arrest. * *p* < 0.05. (**E**) The PBE rates in the control group and Rae1 knockdown group were recorded after 14 h in 2 h GVBD-resumed oocytes after release from IBMX arrest. *p* > 0.05. (**F**) The PBE rates in the control group and Rae1 knockdown group were recorded after 9 h in 2 h GVBD-resumed oocytes after release from IBMX arrest. * *p* < 0.05. The data are presented as mean ± SEM of at least three independent experiments.

**Figure 3 ijms-22-04841-f003:**
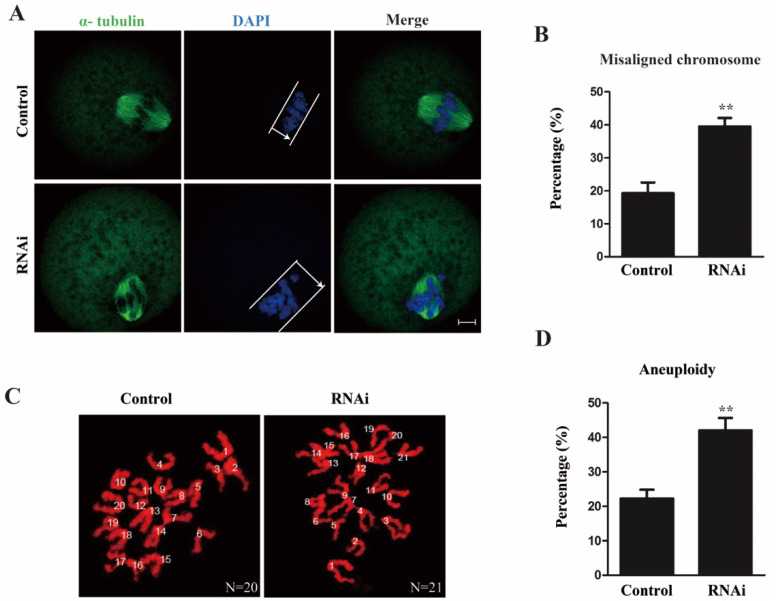
Rae1 knockdown induces abnormal chromosome alignment and elevated aneuploid rate in oocytes. (**A**) Representative confocal images of spindle morphology and chromosome alignment in the control and Rae1 knockdown MI oocytes. The white line and arrow indicate that in the control group, chromosomes are congressed and aligned at the spindle midzone, while in the Rae1-knockdown groups, chromosomes are scattered and dispersed. α-tubulin is shown in green and DAPI is shown in blue. Scale bar = 10 μm. (**B**) The rate of misaligned chromosomes is recorded and compared in the control (*n* = 96) and Rae1 knockdown MI oocytes (*n* = 108). ** *p* < 0.01 (**C**) Representative confocal image of chromosome spread at the MII stage in the control and Rae1 knockdown group. Control oocytes with a normal haploid complement of 20 chromosomes and Rae1 knockdown oocyte with 21 chromosomes. Chromosomes were stained with propidium iodide (PI) and is shown in red. Scale bar = 5 μm. (**D**) The aneuploidy rate is recorded and compared in the control (*n* = 31) and Rae1 knockdown oocytes (*n* = 35). ** *p* < 0.01. The data are presented as mean ± SEM of at least three independent experiments.

**Figure 4 ijms-22-04841-f004:**
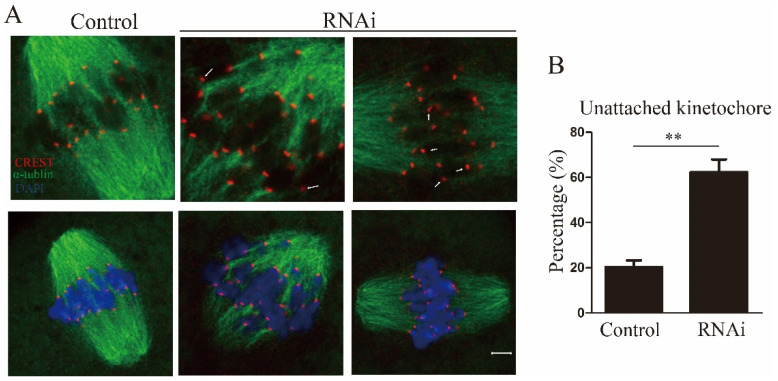
Rae1 knockdown weakens kinetochore–microtubule (K-MT) attachment in oocytes. (**A**) Representative confocal image of K-MT attachment in the control and Rae1 knockdown MI oocytes after cold treatment. α-tubulin is shown in green, CREST in red, and DAPI in blue. White arrows indicate nonconnected kinetochores in the Rae1 knockdown oocytes. Scale bar = 5 μm. (**B**) The rate of unattached kinetochores was recorded and compared in the control (*n* = 125) and Rae1 knockdown oocytes (*n* = 143). ** *p* < 0.01. The data are presented as mean ± SEM of at least three independent experiments.

**Figure 5 ijms-22-04841-f005:**
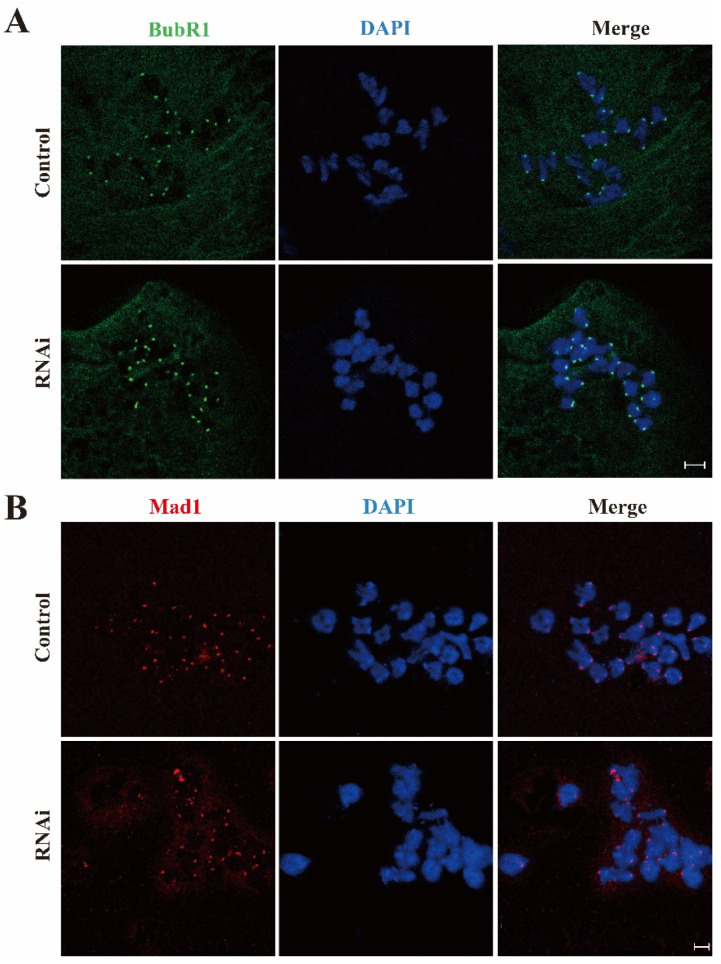
The location of BubR1 and Mad1 was not affected in the Rae1 knockdown oocytes. After Rae1 knockdown, GV stage oocytes were cultured for 8.5 h and then collected for immunofluorescent staining of BubR1 and Mad1. (**A**) Confocal image of BubR1 localization in the control and Rae1 knockdown oocytes by chromosome spreading. BubR1 is shown in green and DAPI in blue. Scale Bar = 5 μm. (**B**) Confocal image of Mad1 localization in the control and Rae1 knockdown oocytes by chromosome spreading. Mad1 is shown in red and DAPI in blue. Scale Bar = 5 μm.

**Figure 6 ijms-22-04841-f006:**
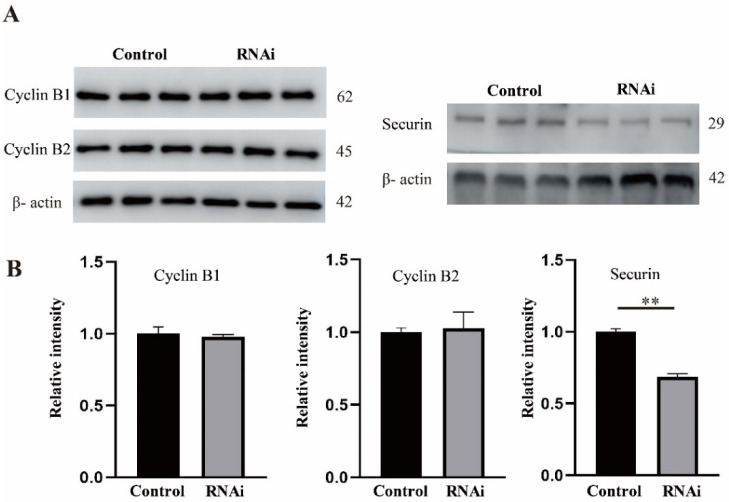
Rae1 knockdown significantly decreases the protein level of securin in MI stage oocytes. After Rae1 knockdown, GV stage oocytes were cultured for 8 h (MI stage) after release from IBMX arrest and were collected for the Western blotting of cyclin B1, cyclin B2, and securin. (**A**) Three independent repeat results of securin, cyclin B1, and cyclin B2 by Western blotting. (**B**) The relative intensity of securin, cyclin B1, and cyclin B2 were compared in the control group and Rae1 knockdown group. ** *p* < 0.01. The data are presented as mean ± SEM of at least three independent experiments.

## Data Availability

The data that support the findings of this study are available from the corresponding author upon reasonable request.

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
