# Peer review of "Ribonucleic Acid Export 1 Is a Kinetochore-Associated Protein That Participates in Chromosome Alignment in Mouse Oocytes"

_ijms, 2021, doi:10.3390/ijms22094841_

Round 1

Reviewer 1 Report

Major comments

In the title: Ribonucleic acid export 1 is not a novel protein. Also, it does not participate in aneuploidy so it is suggested to change the title to:

“Ribonucleic acid export 1 is a kinetochore-associated 2 protein that participates in chromosome alignment in mouse oocytes”.

The immunoblotting experiments that supposed to demonstrate that RAE1 expression levels are highest at the MII stage are not convincing. In panel E, the legend should explain what is presented by the error bars, even though this is presented in the M&M. The error bars are so small that one tends to believe these are technical replicates, not biological. The biological replicates should be presented to the reviewers.

Figure 2 D-F: The percentages and SEM written above the bars is not necessary as these numbers are also mentioned in the text. It makes the figures very busy. Besides, 2 decimals for these percentages digests a level of accuracy that is not realistic. One decimal suffices. Same holds true for Figures 3B,D, and 4B.

Legend Figure 3: It should be mentioned what is represented by the white lines in panel A.

Figure 4, from the pictures it is not evident that K-MT attachment is incorrect. That the chromosomes are not correctly aligned is convincing, but that this is due to abnormal K-MT attachment is not convincing. CREST seems to be normally localized on the chromosomes. This is an overinterpretation of the data.

Line 184: ‘…although the activity of SAC was affected by Rae1 knockdown…’ There is no indication whatsoever that SAC activity was altered after Rae1 knockdown. In fact, the data from figure 5 suggest that SAC activity was not affected.

Figure 6, the immunoblotting data are not convincing. There appear to be other bands present in the Securin blot, therefore the whole blot, not just a fragment, should be presented.

Figure 6, the B-actin and Cyclin B2 blots seem to be indeed from the same blot, but the blots of Securing and Cyclin B1appear to be from a different blot. It should be absolutely clear that these are the same blot and that quantification has been done correctly and independently on at least biologically independent samples

Minor comments

Line 14 nucleoporin instead of Nucleoporins.

Line 14 There is no need to abbreviate (Nups), since this has no function in the abstract.

Line 27 propose instead of proposed.

Line 35 There is no need to abbreviate (Nups), since this abbreviation is not used in the entire text. Please remove.

Line 233 securin, while in line 236 it is capitalized as Securin. Either way is correct as long as it is consequent.

Author Response

We appreciate very much your time and efforts that spent on reviewing our manuscript entitled “Ribonucleic acid export 1 is a kinetochore-associated protein that participates in chromosome alignment in mouse oocytes” (IJMS-1172848). Your opinions have greatly  improved our manuscript, making it more logical and rational.  

Reviewer 2 Report

The presented manuscript is focused on a very interesting topic and is very well written. The methodological approaches used are adequate to the planned experiments and complement each other well.

Author Response

We appreciate very much your time and efforts that spent on reviewing our manuscript entitled “Ribonucleic acid export 1 is a kinetochore-associated protein that participates in chromosome alignment in mouse oocytes” (IJMS-1172848). 

Round 2

Reviewer 1 Report

The manuscript has been improved considerably. There is however one element that is not correct and should be corrected to become eligible for acceptation. This concerns figure 6, where protein expression is presented via immunoblotting. It is clear that these are different blots for the different proteins, as I indicated in my previous review.  Only one actin blot is shown, and the data are quantified. This is not correct, for each blot presented the actin blot should also be presented. In addition, the corresponding actin blot should be used for quantification. This is essential. The way it is presented now is incorrect. For each protein at least 3 different blots should be made (and presented for the reviewers, plus corresponding actin immunos).
